# Technical Note: Comparison of methane ebullition modelling approaches used in terrestrial wetland models

Olli Peltola[1], Maarit Raivonen[1], Xuefei Li[1], Timo Vesala[1,2]

[1]Department of Physics, POBox 68, FI-00014 University of Helsinki, Finland
[2]Department of Forest Sciences, POBox 27, FI-00014 University of Helsinki, Finland

*Correspondence to*: Olli Peltola (olli.peltola@helsinki.fi)

**Abstract.** Emission via bubbling, i.e. ebullition, is one of the main methane ($CH_4$) emission pathways from wetlands to the atmosphere. Direct measurement of gas bubble formation, growth and release in the peat-water matrix is challenging and in consequence these processes are relatively unknown and are coarsely represented in current wetland $CH_4$ emission models. In this study we aimed to evaluate three ebullition modelling approaches and their effect on model performance. This was achieved by implementing the three approaches in one process based $CH_4$ emission model. All the approaches were based on some kind of threshold: either on $CH_4$ pore water concentration (ECT), pressure (EPT) or free-phase gas volume (EBG) threshold. The model was run using four years of data from a boreal sedge fen and the results were compared against eddy covariance measurements of $CH_4$ fluxes.

Modelled annual $CH_4$ emissions were largely unaffected by the different ebullition modelling approaches, however temporal variability of $CH_4$ emissions varied an order of magnitude between the approaches. Hence the ebullition modelling approach drives the temporal variability of modelled $CH_4$ emissions and therefore significantly impacts for instance high-frequency (daily scale) model comparison and calibration against measurements. The modelling approach based on the most recent knowledge of the ebullition process (volume threshold, EBG) agreed the best with the measured fluxes ($R^2=0.63$) and hence produced the most reasonable results, although there was a scale mismatch between the measurements (ecosystem scale with heterogeneous ebullition locations) and model results (single horizontally homogeneous peat column). The approach should be favoured over the two other more widely used ebullition modelling approaches and researchers are encouraged to implement it into their $CH_4$ emission models.

## 1 Introduction

A large fraction of methane ($CH_4$) emitted from wetlands to the atmosphere is released in rapid bubbling events, during which part of the biogenic gas bubbles trapped below the surface are released and transported quickly to the atmosphere. This emission route, called ebullition, has been observed in some wetlands to make a significant contribution to the total $CH_4$ emissions (e.g. Tokida et al., 2007;Yu et al., 2014;Christensen et al., 2003), whereas the rest are emitted more steadily via diffusion or plant-mediated transport (e.g. Le Mer and Roger, 2001). Rapid emission of $CH_4$ in ebullition events allows the emitted $CH_4$ to bypass an oxic zone where the transported $CH_4$ might have been otherwise oxidized prior to reaching the atmosphere (Rosenberry et al., 2006). A characteristic feature of ebullition is that it takes place in sporadic events which are irregularly distributed in space and time (e.g. Klapstein et al., 2014; Tokida et al., 2007). Thus their long-term measurement and quantification in the field has proven to be very challenging and hence a complete understanding of the mechanisms controlling ebullition is still lacking.

Based on the current knowledge an episodic ebullition event takes place when a gaseous-phase gas volume at a certain level below the surface reaches a critical threshold, after which the excess gas is released in an ebullition event. This follows from the fact that for big enough gas bubbles the buoyancy forces exceed the retention forces that have been keeping the entrapped bubbles in place and hence they start to ascend towards the surface (Fechner-Levy and Hemond, 1996). Generally, the critical volumetric threshold for triggering ebullition is considered to be around 10 % (at maximum 20 %) of the total peat volume

(Rosenberry et al., 2006). Based on the existing empirical evidence the released gas bubbles consist largely of $CH_4$ (20…80 % of the bubble gas) (Waddington et al., 2009;Kellner et al., 2006;Tokida et al., 2005;Glaser et al., 2004;Walter et al., 2008) and the rest is nitrogen ($N_2$), oxygen ($O_2$) and carbon dioxide ($CO_2$) (Tokida et al., 2005;Kellner et al., 2006;Walter et al., 2008) which suggest that the bubbles originate from conditions with relatively high pore water $CH_4$ concentrations. The gas

bubbles may form under anaerobic conditions where high pore water concentrations facilitate the formation and build-up of gas-phase bubbles. Existence of bubble formation nuclei is not considered to limit bubble formation.

Considering the ideal gas law and Henry's law, the gas-phase bubble volume can be modified by 1) pressure, 2) temperature or 3) pore water concentration changes (Fechner-Levy and Hemond, 1996). The effect of pressure can be further divided into atmospheric and hydrostatic (i.e. the weight of the water column above) pressure. Hence, if the bubble volume threshold is

also considered, an ebullition event may be triggered by decreasing atmospheric pressure, water table depth (WTD), increasing peat temperature or pore water $CH_4$ concentration. Out of these, decreasing atmospheric pressure has been most often reported to trigger ebullition (Tokida et al., 2007a;Tokida et al., 2005;Waddington et al., 2009;Yu et al., 2014;Strack et al., 2005;Kellner et al., 2006), whereas some studies have also reported the effect temperature (Waddington et al., 2009;Kellner et al., 2006;Yu et al., 2014). Also other forcings (e.g. increasing pressure, wind speed) have been linked with ebullition (Goodrich et al.,

2011;Comas et al., 2011).

According to a recent review by Xu et al. (2016) most (24 out of 40) of the process-based models focusing on $CH_4$ cycling incorporate some kind of an approach to model ebullition. In a seminal modelling paper by Walter and Heimann (2000) a simple approach was adopted: If pore water $CH_4$ concentration exceeded a certain threshold value, then the excess $CH_4$ was directly released to the atmosphere. Thereafter, the approach has been adopted with slight modifications to several $CH_4$ models

and can be regarded as the most widespread method to model ebullition. The most common alteration of the approach is to estimate the concentration threshold based on $CH_4$ solubility (e.g. Wania et al., 2010;Riley et al., 2011).

Pore water concentration threshold was the most widely used ebullition modelling approach in the models reviewed by Xu et al. (2016). It was incorporated in 16 out 24 models (67 %) that included this transport route. However, based on current knowledge this approach can be questioned, since it lacks almost all of the details outlined above about the ebullition process

and hence could result in unrealistically modelled process. Other ebullition modelling approaches have been implemented as well. For instance, Grant (1998), Tang et al. (2010) and Raivonen et al. (2017) triggered ebullition if the total pressure of water-dissolved gases exceeded the ambient pressure at a given depth, whereas in some studies (Segers et al., 2001; Granberg et al., 2001; Zhang et al., 2012) ebullition was modelled using a threshold for gaseous volume at a certain depth below surface. This study was motivated by the fact that in many process-based models ebullition is modelled in a manner which does not

agree with the current knowledge of the process. Hence the models possibly produce erroneous ebullition fluxes and thus may for instance bias the modelled annual $CH_4$ emissions and produce a mismatch between modelled and measured $CH_4$ fluxes. The aim of this study is to compare three ebullition modelling approaches which are based on pore water concentration (Sect. 2.1), pressure (Sect. 2.2) and gaseous volume (Sect. 2.3) thresholds. This is achieved by implementing the three approaches in one process-based model called HIMMELI (Raivonen et al., 2017) and running the model with the same input data and only

altering the ebullition modelling approach. We aim to characterise the differences in 1) amount, timing and depth below peat surface of the modelled ebullition events, 2) variables causing the events and 3) modelled $CH_4$ flux to the atmosphere. In addition, the performance of the volume threshold approach using different model parameters is evaluated. Hypothetically, the approach based on gas volume threshold should produce the most reasonable results since it resembles the current knowledge of ebullition the most. The main aim of this technical note is to report the differences between ebullition modelling approaches

and to promote the usage of physically sound methods in the coming $CH_4$ modelling studies.

## 2 Materials and methods

In this study a model concentrated on $CH_4$ cycling is run using data from a boreal fen to study the differences between three ebullition modelling approaches. The model and measurement data are described below. The following nomenclature is used throughout this study: "Ebullition event" is used to define an episode during which concentration (ECT), pressure (EPT) or bubble volume (EBG) threshold was exceeded at a certain depth, "ebullition" is defined as the upward transport of $CH_4$ via the ebullition process to the lowest air filled pore space and "direct ebullition to the surface" is the ebullition flux directly to the atmosphere. Direct ebullition to the surface can take place only when the water table level is at the surface or above it.

### 2.1 Process-based model HIMMELI

In this study a process-based model HIMMELI provided a framework used to compare the three ebullition modelling approaches. The model is described elsewhere (Raivonen et al., 2017) and thus the description is not repeated in detail here. In short, the 1-D model estimates sources, sinks and interactions between three substances, namely $CO_2$, $O_2$ and $CH_4$ in a vertically layered peat-water-air column. HIMMELI incorporates the following reaction-diffusion equation to model the temporal evolution of $CH_4$ concentration ($c_w$, unit mol m$^{-3}$) in the peat pore water:

$$\frac{\partial c_w(t,z)}{\partial t} = - Q_{diff}(t,z) - Q_{plant}(t,z) - Q_{ebu}(t,z) + R_{prod}(t,z) - R_{oxi}(t,z), \tag{1}$$

where $Q_{diff}$ is the diffusive flux of $CH_4$ in the peat, $Q_{plant}$ is the transport rate of $CH_4$ via plants roots, $Q_{ebu}$ is the transport rate of $CH_4$ via ebullition, $R_{prod}$ is the production rate of $CH_4$ and $R_{oxi}$ is the rate at which oxidation removes $CH_4$ from the pore water. The units for the terms on the right hand side of Eq. (1) are mol m$^{-3}$ s$^{-1}$. In this study we altered $Q_{ebu}$ between the runs, the other terms were not modified.

The model is driven with peat temperature (T), atmospheric pressure, water table depth (WTD), leaf area index (LAI) of gas-transporting vegetation, and rate of anoxic soil respiration. We ran the model for the Siikaneva peatland site (Rinne et al. 2007) using measured T, atmospheric pressure and WTD and simulated LAI and anoxic respiration as input. The latter was simulated similarly to the study by Susiluoto et al. (2017) in which HIMMELI was combined with a model of anoxic respiration. The model simulated respiration as a fraction of NPP plus temperature-dependent peat decay. The former was distributed vertically according to the root distribution while the latter was distributed evenly into the peat layers below water level, therefore, the main factor driving the anoxic respiration was net primary productivity (NPP) modelled for Siikaneva. We used the same calibrated parameter values as in the discussion paper of Raivonen et al. (2017). The model has been calibrated in Susiluoto et al. (2017), however parameters used in the different ebullition modelling approaches (see below) were not calibrated. Instead values found from the literature were used for the ebullition modelling approaches, since this way the results are more comparable across studies (e.g. with Wania et al., 2010). Furthermore, when using literature values the results are truly related to differences between the approaches, and not related to differences between calibrations. In this study HIMMELI was run using a 2 m thick peat column with 10 layers, therefore each layer was 0.2 m thick. The model output time step is one day, but the model uses a shorter internal time step in order to ensure numerical stability.

In HIMMELI, WTD divides the peat column into oxic (air-filled) and anoxic (water-saturated) parts. Anoxic respiration is a source of $CH_4$ and the main part of the anoxic respiration was distributed vertically along an exponential function that describes the vertical distribution of root mass, as root exudates are known to be an important substrate for methanogens (Ström et al., 2003). In contrast to Raivonen et al. (2017) in this study anaerobic decomposition of litter and old peat below water level was also included as a source of $CH_4$. It was modelled with a simple $Q_{10}$-model according to Schuldt et al. (2013).

The model simulates transport of $CH_4$, $O_2$ and $CO_2$ between peat and the atmosphere by diffusion in air-filled and water-saturated peat and through aerenchymatous wetland plant roots, as well as by ebullition. In case ebullition occurs when WTD

is below the peat surface, the ebullited gases are not released directly into the atmosphere but they are transported into the bottom air-filled peat layer. This is called as the lowest air layer throughout the study. The plant transport capacity depends on LAI that determines the root mass available for gas transport. Methanotrophy oxidizing $CH_4$ to $CO_2$ is modelled as a dual-substrate Michaelis-Menten process in which both $CH_4$ and $O_2$ concentrations control the oxidation rate. Consequently, the simulated $O_2$ concentrations affect the $CH_4$ loss rate in the peat.

### 2.1.1 Ebullition based on concentration threshold (ECT)

Walter and Heimann (2000) adopted a simple approach to model ebullition: in their approach if the $CH_4$ concentration at a certain depth exceeded a certain threshold concentration, the excess $CH_4$ was directly transported to the air layer above the water table depth. In this study we follow Wania et al. (2010) and approximate the threshold pore water $CH_4$ concentration to equal the equilibrium concentration calculated based on Henry's law. Thus the rate of dissolved $CH_4$ concentration ($c_w$) change due to ebullition at a certain depth can be calculated as:

$$Q_{ebu} = k \left( c_w - H^{cc} \frac{p_{tot}}{RT} \right), \tag{2}$$

where $k$ is ebullition half-life ($\frac{\ln(2)}{1800\ s}$), $H^{cc}$ is the dimensionless Henry solubility of $CH_4$ calculated based on Sander (2015), $p_{tot}$ is the sum of atmospheric and hydrostatic pressure, i.e. total pressure (Pa), $R$ is the universal gas constant (8.3145 J mol-1 K-1) and $T$ is temperature (K). An equal amount of $CH_4$ that was removed from pore water based on Eq. (2) is immediately released to the lowest air layer which implies that the bubbles ascend fast enough in order to reach the lowest air layer within the model time step.

### 2.1.2 Ebullition based on pressure threshold (EPT)

Tang et al. (2010) criticised the ECT approach since it uses a $CH_4$ concentration threshold to trigger ebullition, whereas in numerous studies decreasing atmospheric pressure has been shown to lead to bubble release event (Green and Baird, 2012; Kellner et al., 2006; Strack et al., 2005; Tokida et al., 2007; Waddington et al., 2009; Yu et al., 2014). Thus they devised a modelling approach which triggers ebullition at a certain depth if the partial pressures of $CH_4$, $CO_2$, $O_2$ and $N_2$ combined exceed the sum of hydrostatic and atmospheric pressure ($p_{tot}$). Tang et al. (2010) took into account also the capillary forces when estimating the pressure threshold, however in this study these forces were neglected. In HIMMELI partial pressures of $CH_4$, $CO_2$ and $O_2$ are explicitly calculated using Henry's law and pore water gas concentrations, whereas partial pressure of $N_2$ is assumed constant (40 % of atmospheric pressure) throughout the peat column. This value for $N_2$ agrees with empirical evidence for instance by Tokida et al. (2005) and Walter et al. (2008). The excess moles are then released directly to the lowest air layer, unlike in Tang et al. (2010) where a re-dissolution of bubbles back to water is allowed during their ascent. The rate of dissolved $CH_4$ concentration change due to ebullition can then be calculated for each layer as:

$$Q_{ebu} = k \frac{f_{ss} c_w}{H^{cc}}, \tag{3}$$

where $f_{ss}$ is the relative supersaturation (dimensionless) calculated as

$$f_{ss} = \begin{cases} \dfrac{p_{tot} - \sum\limits_{i} p_i}{\sum\limits_{i} p_i}, & \sum\limits_{i} p_i \geq p_{tot} \\[4mm] 0, & \sum\limits_{i} p_i < p_{tot} \end{cases} \tag{4}$$

where $p_i$ denotes the partial pressure of $i$th gas calculated based on Henry's law. The same amount of $CH_4$ that was removed from pore water is then released to the lowest air layer. With this approach ebullition of $CH_4$ may take place even though based on Henry's law water is not supersaturated with $CH_4$. Thus ebullition may originate from depths with relatively low pore water
$CH_4$ concentrations when compared to the ECT approach.

**2.1.3 Ebullition based on bubble growth (EBG)**

Fechner-Levy and Hemond (1996) devised a mathematical framework that describes how temperature, pressure and mass transfer to/from a bubble suspended in water alter the bubble volume. They applied this framework to analyse their peatland data. Later Kellner et al. (2006) applied it to model $CH_4$ ebullition and Zhang et al (2012) slightly modified the approach and
implemented it to a larger process-based model (NEST-DNDC). In this approach bubble volumes are calculated and updated constantly based on ideal gas law and Henry's law and if bubble volume at a certain depth exceeds a predefined threshold, then the excess volume is released to the atmosphere. This approach is supported by laboratory and field experiments which have shown that ebullition occurs only if large enough fraction of the overall pore space volume consists of free-phase gas. This fraction, denoted as $f_{V\max}$ in this study, is commonly reported to be approximately 0.1.

Initially a bubble volume is formed at a certain depth if $CH_4$ concentration exceeds the concentration that the water can withhold based on Henry's law and assuming that the bubble $CH_4$ mixing ratio ( $c$ ) is fixed at 50 % (see Table 1). The excess $CH_4$ is transferred into a gaseous volume calculated based on the amount of excess moles and the predefined bubble $CH_4$ mixing ratio. At each model layer this volume is divided evenly between $N_{bub}$ spherical bubbles. The bubbles do not have any interaction between each other (no coalescence etc) and they remain stationary in the peat-water matrix. In principle $N_{bub}$ is
merely a tuning parameter which controls the rate of mass flux between the gas volume and the pore water. Once the bubbles have been formed the $CH_4$ exchange between the stationary bubbles and the pore water is calculated based on Epstein and Plesset (1950):

$$Q_{ebu} = \frac{4\pi r D N_{bub}}{V_w} \left( c_w - \frac{H^{cc} c p_{tot}}{RT} \right), \tag{5}$$

where $r$ is the radius of one bubble (m), $D$ is temperature dependent effective diffusion coefficient of $CH_4$ in water (m$^2$ s$^{-1}$)
and $V_w$ is the volume of pore water in this model layer (m$^3$). $D$ was calculated based on Arah and Stephen (1998) and in order to take into account the fact that the media did not consist solely of water, but it was a peat-water mixture, the calculated value for $D$ was multiplied with 0.9 prior usage (Raivonen et al., 2017). For simplicity, temperature and pressure inside the bubble volume were assumed to be equal to their pore water counterparts and the mass transfer was assumed to be stationary within the model time step. In order to keep the model in balance the rate of bubble $CH_4$ ( $n_b$ , unit mol) change at a specific
model layer is

$$\frac{\partial n_b}{\partial t} = -Q_{ebu} V_w \tag{6}$$

Thus in this modelling approach $CH_4$ can be also transferred from the bubbles back to the pore water surrounding the bubbles. This kind of feedback is missing from the other ebullition modelling approaches used in this study.

The bubble volume is updated after every model time step based on Eq. (7):

$$\Delta V = \frac{\Delta n_b}{c_b} + \frac{\Delta T}{T} V - \frac{\Delta p_{tot}}{p_{tot}} V ,$$  (7)

where $V$ is the combined volume ($m^3$) of all bubbles at a specific model layer and $c_b$ is the bubble $CH_4$ concentration (mol $m^{-3}$). The terms on the right hand side of Eq. (7) represent the change in the volume due to diffusion of $CH_4$ to/from the bubbles (cf. Eq. (6) above), change in volume due to temperature or total pressure change. From now on they are called as the c-, T- and p-term, respectively. The T- and p-terms can be readily calculated from the model input data, whereas the c-term can be determined based on Eq. (6).

As stated before, an ebullition event occurs only if bubble volume at a certain depth exceeds a predefined threshold ($V_{max}$):

$$F_{ebu} = \begin{cases} 0, & V + \Delta V \le V_{max} \\ c_b \dfrac{V + \Delta V - V_{max}}{\Delta t}, & V + \Delta V > V_{max} \end{cases} ,$$  (8)

where $F_{ebu}$ is the ebullition flux of $CH_4$ to the lowest air layer (mol $s^{-1}$), $V_{max}$ can be calculated as a product of $V_w$ and $f_{V max}$ and $\Delta t$ is the model time step. Finally the updated bubble volume at a certain depth is

$$V = \begin{cases} V + \Delta V, & V + \Delta V \le V_{max} \\ V_{max}, & V + \Delta V > V_{max} \end{cases} ,$$  (9)

and the amount of $CH_4$ moles in the bubbles at a certain model layer after each model time step is

$$n_b = \frac{c p_{tot} V}{RT}$$  (10)

In order to take into account bubble movement in the peat column after it has been released (i.e. $V + \Delta V > V_{max}$) a simple approach was adopted: excess bubble volumes are released starting from the bottom of the peat column and while the gaseous volume is ascending it will get stuck at certain 0.2 m thick model layer with a probability of $P$, which was set to 0.3. Thus

for instance bubble released from 1 m below the lowest air layer will reach the air layer with a probability of 0.17 ( $(1 - 0.3)^5 \approx 0.17$ ). Otherwise it will stay at the depth where it got stuck and its volume will be updated during the next time step with the procedure described above. This process is repeated for each layer where $V + \Delta V > V_{max}$ and thus at the end of the time step bubble volumes are always smaller or equal to $V_{max}$. This kind of approach produces somewhat similar bubble movement as the "inverted bubble avalanches" modelled with the approach suggested by Coutlhard et al. (2009) and used by

Ramirez et al. (2015). By setting the probability $P$ to 0 the released excess bubble volume will always reach the lowest air layer, similarly as with the other ebullition modelling approaches included in this study. The performance of the EBG approach using different values for $P$ and other parameters are evaluated in Sect. 3.4.

The effect of changing water table level was taken into account in the following way: if the water table dropped below a layer which contained bubbles, the $CH_4$ in the bubbles was immediately released to the newly formed air filled pore space. If the

water table rose, the new water clogged pore space did not initially contain any bubbles.

## 2.2 Site and measurements

The measurements were carried out at an oligotrophic open fen part of the Siikaneva wetland complex which is situated in the southern part of Finland ($61°50'N, 24°12'E$, 162 m a.s.l.). The site is in the boreal region with an annual average temperature of $3.3°C$ and rainfall of 710 mm (Drebs et al., 2002). The vegetation composition is dominated by sedges (*C.*
*rostrata, C. limosa, E. vaginatum*), Rannoch-rush (*Scheuchzeria palustris*) and peat mosses (*Sphagnum balticum, S. majus, S. papillosum*). Peat depth at the site varies between 2 and 4 meters. See more details in Riutta et al. (2007) and Rinne et al. (2007).

The HIMMELI model was driven using measured peat temperature, atmospheric pressure, WTD and simulated LAI and anoxic respiration. See the details about LAI and anoxic respiration simulation in Raivonen et al. (2017) and Susiluoto et al. (2017).
Peat temperature was measured at five depths below the surface (-5 cm, -10 cm, -20 cm, -35 cm and -50 cm) and the measurements were interpolated linearly in time and space in order to match every model time step and depth. Temperatures below -50 cm were obtained by assuming that at 3 m the peat temperature is constant ($7°C$, average temperature at -50 cm). Also WTD and atmospheric pressure time series were gap-filled using linear interpolation. Air pressure was not measured at the site and hence measurements made 5 km away at SMEAR II (Hari and Kulmala, 2005) site were used.

In this study the modelled daily $CH_4$ fluxes are compared against fluxes obtained with the eddy covariance (EC) method at the Siikaneva site. Data from years 2008…2011 are used. EC measurements estimate the ecosystem scale emissions of measured compounds ($CH_4$ in this study) and hence the measured fluxes integrate all the three $CH_4$ emission pathways (plant transport, diffusion and ebullition) at ecosystem scale. The EC setup consisted of a sonic anemometer (USA-1, METEK GmbH) which measured the three wind components and air temperature and a fast response gas analyser used to measure $CH_4$. The
instruments were measuring the turbulent fluctuations 2.75 m above the peat surface. There were some changes in the $CH_4$ instrumentation during the years. The $CH_4$ analysers used were RMT-200 by Los Gatos Research (2008…2011), TGA-100 by Campbell Scientific (04/2010…08/2010) and G1301-f by Picarro (04/2010…10/2011). The gas analysers were located in a separate housing which protected the instruments from the outdoor conditions. A sampling line consisting of filters and heated Teflon tubing was used to sample air to the closed-path gas analysers. See more details in Peltola et al. (2013).

The EC measurements were made at 10 Hz from which the fluxes were calculated as a covariance between vertical wind and gas concentration using 30-min averaging time. Coordinates were rotated with sector-wise planar fitting and high frequency losses were corrected using empirical procedures (cf. Peltola et al., 2013). All these EC data post-processing steps were done using EddyUH (Mammarella et al., 2016). The EC data were carefully processed keeping in mind the nature of the ebullition process (rapid release events with significantly elevated concentrations) and processing steps such as de-spiking were tuned
so that these events were not flagged as erroneous measurements.

## 3 Results

### 3.1 Timing and depth of ebullition events

The EPT approach resulted in the highest total amount of ebullition events (866 events), followed by EBG (797) and ECT (389) (Fig. 1). An ebullition event is defined as a time period and depth where concentration (ECT approach), pressure (EPT)
or volume (EBG) threshold was exceeded (cf. Sect. 2). As stated in Sect. 2.2 EPT approach does not require that the pore water $CH_4$ concentration reaches supersaturation, it uses the total pressure of water-dissolved gases to trigger an ebullition event, which is a less strict requirement for ebullition. Hence higher amount of events were observed with EPT approach. ECT approach requires high pore water $CH_4$ concentrations in order to trigger an ebullition event which limits the amount of ebullition events, whereas EBG may trigger an ebullition event due to four reasons: increased $CH_4$ concentration in pore water
(c-term of Eq. (7)), increased temperature (T-term of Eq. (7)), decreased WTD or atmospheric pressure (p-term of Eq. (7)).

Most of the events took place in July, for ECT, EPT and EBG approaches 43 %, 35 % and 36 % of all events happened during that particular month, respectively (Fig. 1). The vertical distributions of the ebullition events in Fig. 1 show that for EBG and EPT approaches the events usually originated from below 1 m depth and for ECT below 1.4 m depth. For EBG this means that below 1 m depth the conditions were favourable for bubble volume increase which lead the volume to exceed the maximum volume allowed (Vmax) and hence to trigger an ebullition event. In order to sustain and grow bubbles with the EBG approach relatively high pore water $CH_4$ concentrations are needed (cf. Eq. (5)) and thus the ebullition events originated from depths below rooting depths and layers with oxic pore water (Fig. 1d). For the same reason ECT approach resulted in ebullition events only from the deep pore water. $CH_4$ in the pore water below rooting depth resulted mostly from decay of old peat, since root exudates were not present and if this source of $CH_4$ was omitted in the model, all the approaches stopped producing ebullition events altogether.

Less than half (41 %) of the ebullition events triggered by the EBG approach coincided with a co-located event triggered by the ECT approach, whereas 78 % of the events took place at the same depth and time as with the EPT approach. Most (70 % and 78 %, respectively) of the events happened at the same time as the events modelled with the ECT and EPT approaches regardless of depth. On the other hand, 72 % of the events triggered with the EPT approach matched EBG events, while they matched only 43 % of the ECT events. Hence EPT and EBG triggered ebullition events mostly at the same time and location, however ECT differed more from the other two approaches.

Decreasing WTD (i.e. decreasing hydrostatic pressure) and air pressure triggered approximately 66 % and 67 % of ebullition events when using the EPT approach, respectively. Almost all (94 %) of the ebullition events observed with EPT approach coincided with either decreasing WTD or decreasing atmospheric pressure, the rest 6 % were triggered solely by increasing partial pressures, i.e. increase in $CH_4$, $CO_2$ or $O_2$ pore water concentrations (cf. Sect. 2.2). For EBG approach large proportion of the events were triggered simultaneously due to all the three terms, namely the c-, p- and the T-term (47 %), only 1 % of the events were caused solely by decreasing total pressure, i.e. the p-term. Individually examined, almost all of the events took place due to the c-term (99 %) or the p-term of Eq. (7) (86 %), whereas the T-term had slightly smaller impact (57 % of all events). Most (72 %) of the ebullition events triggered by the p-term coincided with decreasing atmospheric pressure and 59 % with decreasing WTD.

## 3.2 $CH_4$ and bubble volume profiles and dynamics

EPT approach showed the lowest pore water $CH_4$ concentrations (0.39 mol m$^{-3}$), EBG and ECT calculated on average 1.7 and 2.5 times higher values (0.66 and 0.98 mol m$^{-3}$, respectively) and the difference was emphasized at the deepest layers (cf. Fig. 2). If the concentrations were converted to gaseous phase partial pressures using Henry's law and scaled with the total pressure ($p_{CH4}/p_{tot}$), the reason for the differences could be found. With EBG approach the diffusive mass transfer between the bubbles and the pore water was directed from the water to the bubbles if $p_{CH4}/p_{tot}$ exceeded the predefined bubble $CH_4$ mixing ratio $c$ which was set to 50 % in this study. Conversely, the $CH_4$ flux was directed from the bubbles to the pore water if the ratio was smaller than $c$. This follows directly from the Eq. (5), more specifically from the difference between $c_w$ and the equilibrium concentration ($\dfrac{H^{cc} c p_{tot}}{RT}$). Hence, the $CH_4$ concentrations rose until $p_{CH4}/p_{tot}$ exceeded $c$ after which part of the produced $CH_4$ went into the bubbles instead of staying in the pore water, which limited the increase of the $CH_4$ pore water concentrations. If the effect of surface tension would have been included in the pressure inside the bubbles, higher $p_{CH4}/p_{tot}$ values would have been needed (i.e. higher pore water $CH_4$ concentrations) to transfer $CH_4$ from the pore water to the bubbles. The build-up of pore water $CH_4$ concentrations took place only below 1 m depth since closer to the surface the plant roots effectively removed $CH_4$ from the pore water via aerenchyma (compare Figs. 1d and 3). For ECT approach the pore water $CH_4$ concentrations increased until $p_{CH4}/p_{tot}$ equalled unity (i.e. supersaturation of $CH_4$ in pore water) after which the excess

CH$_4$ was removed in ebullition events. This implies that with ECT approach the bubbles consist 100 % of CH$_4$. On the other hand, for EPT approach the increase in pore water CH$_4$ concentrations were limited by the fact that the sum of CH$_4$, CO$_2$, N$_2$ and O$_2$ partial pressures was allowed to be at maximum p$_{tot}$ and since p$_{O2}$ values were low and p$_{N2}$ was 40 % of the atmospheric pressure throughout the peat column, then the sum of p$_{CO2}$ and p$_{CH4}$ below the rooting depth could be around 50…65 % of p$_{tot}$, depending on depth. Hence the low pore water CH$_4$ concentrations in Fig. 2c. p$_{CH4}$/p$_{tot}$ (i.e. the bubble CH$_4$ mixing ratio) during ebullition events with the EPT approach was generally between 25 % and 38 %, whereas p$_{CO2}$/p$_{tot}$ (i.e. the bubble CO$_2$ mixing ratio) was between 28 % and 37 %.

Bubble volumes modelled with the EBG approach mostly resided below 1 m depth (Fig. 3) where the conditions were favourable for bubble growth. Bubble volumes were affected by three terms: CH$_4$ transfer between the bubbles and pore water (c-term) and expansion/contraction due to temperature (T-term) or pressure changes (p-term). Also input of gaseous volume released from deeper layers increased the bubble volumes, whereas bubble release limited the volume to be at maximum V$_{max}$ (Fig.4).

The c-term showed clear seasonality (decreasing bubble volumes during late summer and autumn and increasing volumes in spring and early summer), since it was closely related to temperature (cf. Eq. (5)): CH$_4$ solubility decreases with temperature (i.e. H$^{cc}$ decreases) and due to enhanced CH$_4$ production, pore water CH$_4$ concentrations (c$_w$) increase with temperature. Bubble volumes released from deeper layers and which were re-attached during their ascent maintained the gas volume at -1.1 m depth even though the c-term on average decreased the volume (cf. Fig 4a).

The T- and p-terms generated only temporal variation in the volumes without any permanent increase/decrease. The T-term generated seasonality to the bubble volumes, although mostly the seasonality was controlled by the c-term. The p-term caused strong short-term variation and a small seasonal cycle due to the annual cycle of WTD. Altogether the combination of all these terms, in addition to the input and release of gaseous volume led the bubble volumes to reach their maxima in July and minima in April and generated the ebullition event profile and seasonality shown in Fig. 1a.

### 3.3 CH$_4$ emissions to the atmosphere

The EPT approach resulted in significantly higher temporal variability in the CH$_4$ emission to the atmosphere than the other ebullition modelling approaches (Fig. 5) due to the fact that it produced more ebullition events which caused the short term variability in the CH$_4$ emissions. In order to quantify the short term variability, the amount of variance in the time series at high frequencies were estimated by calculating the time series' power spectral densities and integrating them over the frequency range of interest. Variability at shorter than one week time scale contributed approximately 24, 12 and 7 % to the total time series variance obtained with the EPT, ECT and EBG approaches, respectively, whereas for the measured CH$_4$ flux time series the contribution was 3 % to the total variance. There were also differences in the total time series variance: EPT and ECT gave 92 and 17 % higher variances for the CH$_4$ flux time series than the measurements, whereas EBG estimated 13 % lower variance for the CH$_4$ emission time series. These results can be qualitatively observed in the Fig. 5: the EPT approach showed clearly highest variability, especially at short time scales, followed by ECT and finally EBG and the measured CH$_4$ flux time series. EBG approach explained the variability in the measured CH$_4$ emissions the best (R$^2$ was 0.63), followed by ECT (0.56) and EPT (0.35). These results are consistent with the differences in time series variance outlined above. However, the model results represent CH$_4$ exchange from a single horizontally homogeneous peat column, whereas the measured fluxes correspond to ecosystem scale CH$_4$ exchange. Hence they are not fully comparable with each other.

Even though the CH$_4$ flux time series variances were affected by the ebullition modelling approach, the annual net CH$_4$ emissions were not largely different between the three modelling approaches (Table 2). At maximum the modelling approaches diverged in year 2010 by 1.0 g(CH$_4$) m$^{-2}$ yr$^{-1}$ (8 % of the annual emission). In general all the approaches showed similar decreasing trend from year 2008 to 2010 in the annual CH$_4$ emissions and none of them showed always the lowest or highest emissions. Similar annual CH$_4$ emissions could have been expected given the fact that ebullition does not directly alter the

CH$_4$ production and hence it only provides another transport pathway for emissions which would happen at annual time scale regardless of the transport route, although CH$_4$ oxidation in the peat column might be different between the modelling approaches.

The relative magnitude of different emission pathways varied between the modelling approaches (Fig. 5b, 5d and 5f and Table 2). On annual scale for EBG approach the CH$_4$ emissions via plant aerenchyma contributed approximately 90 and diffusion 10 % of the total CH$_4$ emissions, for ECT the percentages were 80 and 20 % and for EPT 60 and 40 %, respectively. In HIMMELI the ebullition flux is released to the lowest air layer which is often below peat surface and hence the diffusion flux to the atmosphere contains also the ebullition flux signal. Due to the same reason the direct ebullition to the surface is rare (Fig. 5), since ebullition events usually take place when WTD is below surface and thus ebullition flux is not released directly to the atmosphere. The ebullition flux to the lowest air layer was 4 to 9 times higher in EPT approach than in EBG approach and also approximately double that of calculated with the ECT approach (Table 2) which was related to the overall higher amount of ebullition events modelled by the EPT approach (see Sect. 3.1). In addition with EBG approach many of the ebullition events were stuck during their ascent and then dissolved back to the pore water and thus they did not reach the lowest air layer and contribute to the ebullition flux.

## 3.4 Testing the EBG approach with different model parameter values

The EBG approach was tested with different parameter values (cf. Table 1) in order to evaluate the sensitivity of the results on the used parameters. This analysis revealed that the EBG approach was the most sensitive to the bubble CH$_4$ mixing ratio. If the mixing ratio was set to 20 % (parameter set 2.1), instead of the 50 % used in the default run, 2164 events were triggered with the EBG approach which is approximately three times the amount observed with the default run (cf. Fig. 6b). Consequently, more CH$_4$ was transported to the lowest air layer via ebullition (annual mean: 1.2 g(CH$_4$) m$^{-2}$ yr$^{-1}$, with default run: 0.7 g(CH$_4$) m$^{-2}$ yr$^{-1}$, Fig. 6d) and hence slightly larger fraction of CH$_4$ was emitted to the atmosphere via diffusion (14 %, Fig. 6e). However, the total amount of emitted CH$_4$ to the atmosphere was approximately 5 % lower (Fig. 6c), possibly due to the fact that the large amount of ebullition events transported CH$_4$ to upper oxic peat layers where the CH$_4$ was oxidized rather than released to the atmosphere. Also, bubbles were formed closer to the surface than in the EBG default run (cf. Fig. 3 for the default run). These results were reasonable given the fact that the lower mixing ratio facilitated the CH$_4$ transfer from the pore water to the bubbles and hence the bubble growth. On the other hand, higher bubble CH$_4$ mixing ratio (parameter set 2.2) decreased the amount of ebullition events to 394 (50 % decrease) and limited the bubble formation to depths below 1.2 m, which was deeper than in the default run (Fig. 3).

The variance of the modelled CH4 flux to the atmosphere was the most sensitive to the probability that the released bubble volume will get stuck while it is ascending (Fig. 6a). If the probability was increased from the default value (parameter set 4.2), the flux time series variance was slightly decreased. If the probability was set to 0 (parameter set 4.1, cf. Table 1), the EBG results resembled results obtained with the ECT approach since the ebullition flux was more directly linked with bubble production which was in turn driven by changes in CH$_4$ pore water concentration (cf. Fig. 4). Ebullition flux to the lowest air layer was approximately three times higher (Fig. 6d), since all the ebullition events reached the layer and were not stuck during their ascent. This also lead the diffusion pathway to have a higher fraction of the overall emission of CH4 to the atmosphere (28 % vs. 8 % with the default run, Fig. 6e).

If the threshold fraction for gaseous volume ($f_{Vmax}$) was decreased to 5 % (parameter set 1.1, cf. Table 1) the amount of ebullition events was increased by 59 % (1268 events observed), since smaller bubble volumes were needed before an ebullition event was triggered. Larger value for $f_{Vmax}$ (parameter set 1.2) decreased the amount of ebullition events by 26 % (Fig. 6b). Otherwise changing the value for $f_{Vmax}$ had a negligible impact on the model results. Increasing/decreasing the amount of bubbles in a model layer ($N_{bub}$) accelerated/decelerated the CH$_4$ transfer between the bubbles and pore water (cf.

c-term in Eq. 4), but on the whole it had a minimal effect on the modelled ebullition (see the results for parameter sets 3.1 and 3.2 in Fig. 6).

**4 Discussion**

All of the modelling approaches produced ebullition events only deep below the surface. Hence the results resemble the "deep peat" hypothesis put forward by Glaser et al. (2004) in which the free-phase gas is produced in deep (> 1 m) peat and trapped under semiconfined layers, even though peat micro structure (e.g. woody layers) was not described in the model.. According to the hypothesis, these layers episodically rupture due to changes in gas volume buoyancy inflicted by for instance pressure changes. Hence it links ebullition mostly to the processes that take place in the deep peat and it has been supported by some field studies. For instance in a relatively recent study Bon et al. (2014) observed high $CH_4$ pore water concentrations below 2 m depth which they claim to be an indication of free-phase gas and hence ebullition from the deep peat.

There is, however, mounting evidence that bubble formation and release is also taking place close to the surface, which is in contrast to the "deep peat" hypothesis. Hence, Coulthard et al. (2009) argued that the bubble formation and ebullition is more directly linked with processes that take place close to the surface, primarily because most of the labile fresh carbon is located in the rooting zone and hence $CH_4$ production is higher in the shallow peat than deeper below the surface. The "shallow peat" hypothesis was supported for instance by Klapstein et al. (2014) who showed that in their field study over 90 % of ebullition occurred in the surface peat layer and the carbon in the bubble $CH_4$ was recently fixed from the atmosphere. As mentioned before for the EBG modelling approach pore water $CH_4$ concentrations need to be high enough (above $\dfrac{H^{cc}cp_{tot}}{RT}$) to form and grow a bubble at a certain model layer. This sets a strict limit where bubbles may exist and the presence of vascular plants' roots effectively prevented bubble formation close to the surface by limiting the pore water $CH_4$ concentrations. Thus there is clearly a conflict between the "shallow peat" hypothesis and the EBG modelling approach. It could be slightly alleviated by lowering the bubble $CH_4$ mixing ratio ( $c$ ), which would allow bubbles to exist at lower pore water $CH_4$ concentrations, or by using a predefined profile for $c$ (increase with depth), instead of one constant fixed value. In general the reported bubble $CH_4$ mixing ratios are at minimum 10 %, which equals at $T = 15°C$ and $p = 1013 \text{hPa}$ based on Henry's law equilibrium $CH_4$ pore water concentration of 0.15 mol m$^{-3}$. However, for instance Baird et al. (2004) have shown bubble build-up initiation at approximately 5-times lower average $CH_4$ pore water concentrations. This could be explained by strong small scale variability in pore water $CH_4$ concentration, which is undetectable with the current measurement methods and hence by the methods used by Baird et al. (2004). Such variability could create small "pockets" of high $CH_4$ concentrations where bubble formation and growth could take place. That kind of variability cannot be however readily implemented in 1-D column models and alternate ways of dealing with this issue should be developed.

It is also possible that the vascular plants in reality did hinder bubble formation and growth at the study site, deeming the EBG results plausible. However, this cannot be confirmed since free-phase gas content was not measured at the site. Coulthard et al. (2009) claim that vascular plants do not necessarily inhibit ebullition from shallow peat due to their strong influence on $CH_4$ production, whereas Chanton et al. (2005) suggest the opposite. Also the empirical studies are not conclusive in this matter. For instance Klapstein et al. (2014) did not observe any negative effect of sedge cover on ebullition, whereas Green and Baird (2011) showed that the percentage of $CH_4$ emitted in episodic ebullition events decreased with the presence of sedges. Green and Baird (2011) conclude that the sedge effects on $CH_4$ emissions via ebullition may be species dependent. Clearly the effect of sedges on ebullition should be studied more prior to these effects can be implemented in terrestrial CH4 models.

While the bubble movement was implemented in the EBG approach in a relatively simple way note that in most of the modelling studies it has been ignored completely. Ramirez et al. (2016) showed that the peat pore structure had a significant influence on the bubble size distribution and release and their results suggest that peat structure might be more important than production rate in controlling ebullition. Hence emphasis on future ebullition modelling should be on describing the bubble movement in a simplistic, yet accurate way. The reduced complexity model MEGA developed by Ramirez et al. (2015) is a step in the right direction in this respect. In the EBG approach used in this study the effect of peat pore structure on bubble movement can be controlled by modifying the probability at which bubbles adhere at certain levels while they ascent ($P$) and by changing the volume threshold ($V_{max}$) after which the bubbles are released. A profile for both of these parameters would allow to take into account the vertical variation in peat pore structure and its effect on bubble mobility and accumulation (Chen and Slater, 2015).

Despite the obvious shortcomings of the EBG approach discussed above it still produced the best match against measured $CH_4$ fluxes with a relatively high coefficient of determination ($R^2=0.63$). However, the EC system estimated fluxes at ecosystem scale whereas the model results represent fluxes from a single horizontally homogeneous peat column and hence they are not fully comparable. The EC source area (i.e. footprint) may contain locations where ebullition is taking place and locations where at the same it is absent. Therefore, the ebullition events are presumably averaged with other sources (e.g. plant transport) in the conventionally processed EC fluxes, yet their impact on ecosystem scale fluxes is captured accurately. Hence, while EC does estimate fluxes on a continuous basis, EC derived fluxes are not a perfect measure of bubble flux from a single peat column. Ideally these kind of models should be tested against autochambers that are continuously estimating the $CH_4$ flux at a smaller horizontal scale than EC. Moreover, ebullition events can be detected and quantified from such measurements (e.g. Goodrich et al., 2011) and hence could be directly compared against the modelled ebullition. Besides autochambers, the models could also be validated against changes in $CH_4$ storage (liquid and gas phase) in the peat, since ebullition modelling approach largely alters the amount of $CH_4$ stored below the surface during the model run (cf. Fig. 2).

The process-based models are often evaluated by comparing with measurements and possibly optimised to match observed $CH_4$ emissions by minimising some statistic or objective function, for instance root mean square error (RMSE) (Wania et al., 2010) or Nash-Sutcliffe efficiency (NS) (van Huissteden et al., 2009), or by using for instance Markov chain Monte Carlo methods (Susiluoto et al. 2017). Poorly represented processes, for instance ebullition, hinder this comparison and may yield erroneous values for many of the model parameters when the models are calibrated, even though other processes would be described in a realistic manner. This is simply because the poorly described process causes apparent mismatch between models and measurements. This might be especially true for ebullition, since it strongly influences $CH_4$ flux time series variance which in turn has a direct impact on many metrics, such as RMSE, NS and $R^2$. Hence, slightly more realistic ebullition modelling, such as the EBG approach, would allow better tuning of the $CH_4$ models and ultimately more accurate $CH_4$ emission estimates. Finally it should be mentioned that there is also merit in simplicity. While the ECT approach lacks feedback to many ebullition drivers (e.g. pressure changes), the approach is simple and hence adds only a minimal amount of degrees of freedom to the model and therefore possibly provides more robust modelling results. The approach could be further modified to take into account the fact that bubbles do not consist 100 % of $CH_4$ (e.g. Riley et al., 2011) in order to make it more realistic. On the other hand the merit of the EPT approach, besides being a simple approach, is that it does not use a predefined $CH_4$ mixing ratio in the bubbles and hence it is a viable modelling approach for models which explicitly calculate also $CO_2$ and $O_2$ concentrations in the peat column.

**5 Conclusions**

In this study three approaches to model $CH_4$ transport via bubbling, i.e. ebullition, were compared by implementing them in one peatland $CH_4$ cycling model called HIMMELI. The model was run using forcing data from a boreal sedge fen. The study

was motivated by the fact that ebullition is modelled rather crudely in many models and hence there is clearly a need for improvement and comparison of methods. All the three approaches were based on thresholding on some variable, either pore water $CH_4$ concentration (ECT approach), pressure (EPT) or free-phase gas volume (EBG). The ECT approach is commonly used in process-based $CH_4$ models even though it describes the physical processes behind ebullition in a simplistic manner, whereas EBG approach resembles most closely the current knowledge on the process.

EPT simulated the highest amount of ebullition events and hence produced the highest ebullitive $CH_4$ fluxes to the surface. All the three modelling approaches triggered ebullition events only well below the surface, which was caused by the effect of vascular plant root distribution on pore water $CH_4$ concentrations. The modelled $CH_4$ fluxes were also compared against eddy covariance (EC) measurements of $CH_4$ fluxes and EBG produced the best match against measurements ($R^2=0.63$), although a horizontal scale mismatch is evident between the model results (single peat column) and EC measurements (ecosystem scale). EBG incorporates most of the ebullition drivers observed in different studies (temperature, pressure, $CH_4$ production, water level changes), whereas the other modelling approaches, especially ECT, are missing a link to many of the drivers listed. While simple modelling approach, such as ECT, may yield robust results without many tunable parameters, overly simplified processes in models may hinder model comparison against measurements. Hence modellers and researchers are encouraged to incorporate a realistic description of the ebullition pathway to their models.

### Code availability

The Fortran codes describing the EBG modelling approach are included as supplementary material to this article.

### Competing interests

The authors declare that they have no conflict of interest.

### Acknowledgements

We want to acknowledge all the researchers who have contributed in the development of the HIMMELI model and also researchers and technical staff who have continuously maintained measurements at the Siikaneva peatland site. The study was supported by the National Centre of Excellence (272041), ICOS-Finland (281255), CARB-ARC (286190) and Academy professor project (284701) funded by the Academy of Finland and AtMath funded by University of Helsinki.

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

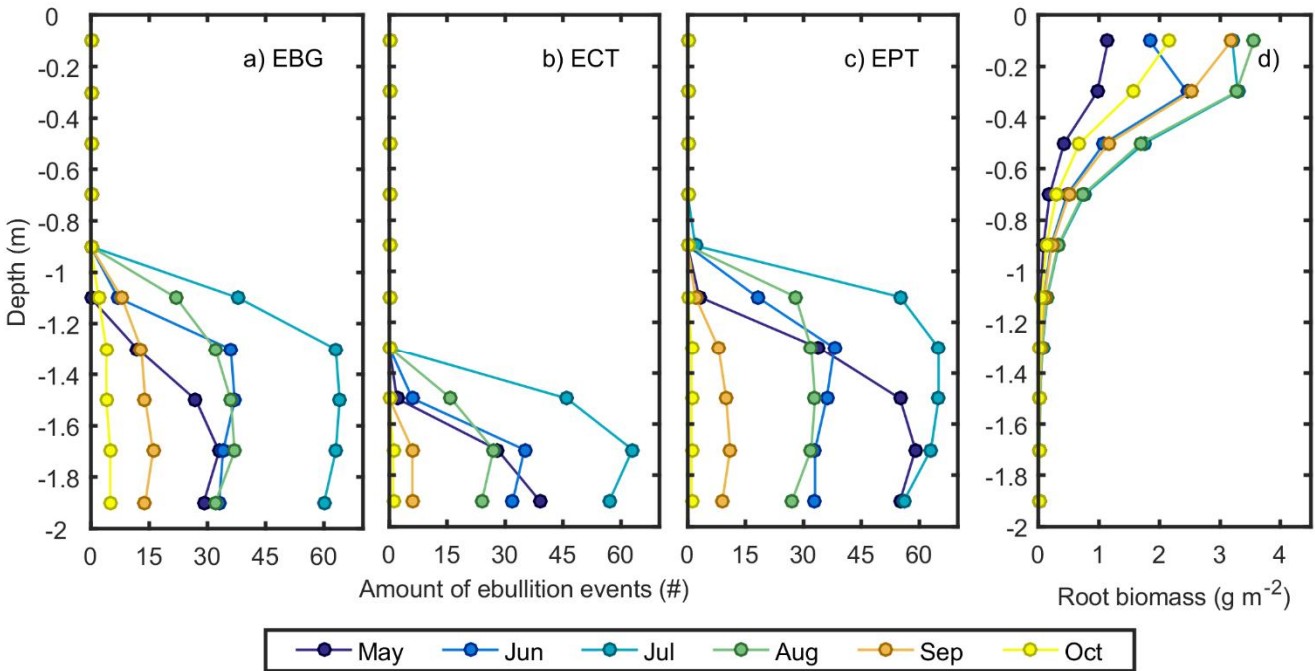

**Figure 1: Amount of ebullition events at different months and depths (subplots a…c) and distribution of plant roots under water table (subplot d). During other months (November…April) there were only 1 % of all ebullition events and thus data from those months are not shown.**

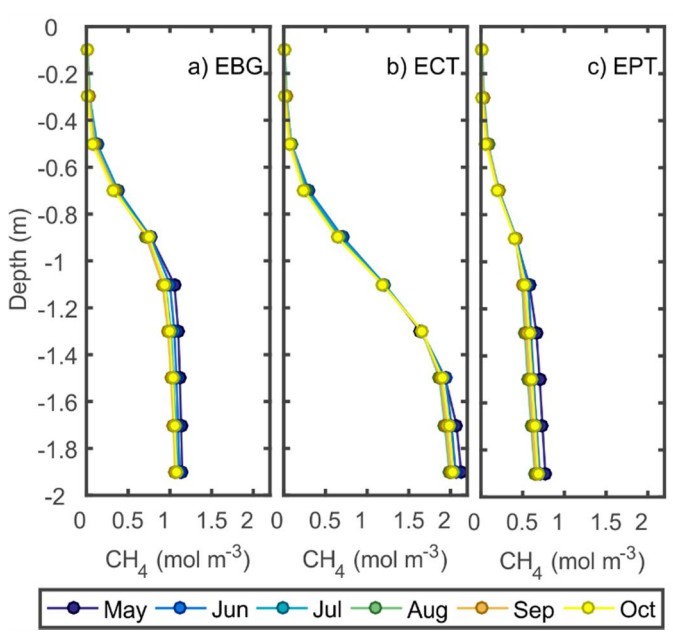

**Figure 2: Average CH₄ pore water concentrations at different depths and months.**

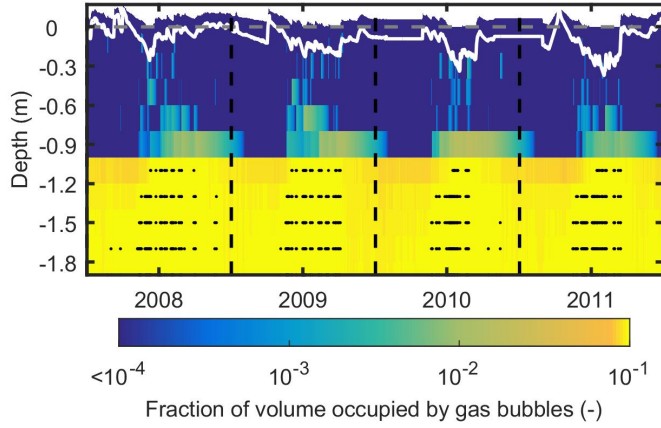

**Figure 3: Bubble volume profiles calculated with the EBG approach. Black dots highlight periods and depths with ebullition events. White line shows the water table depth.**

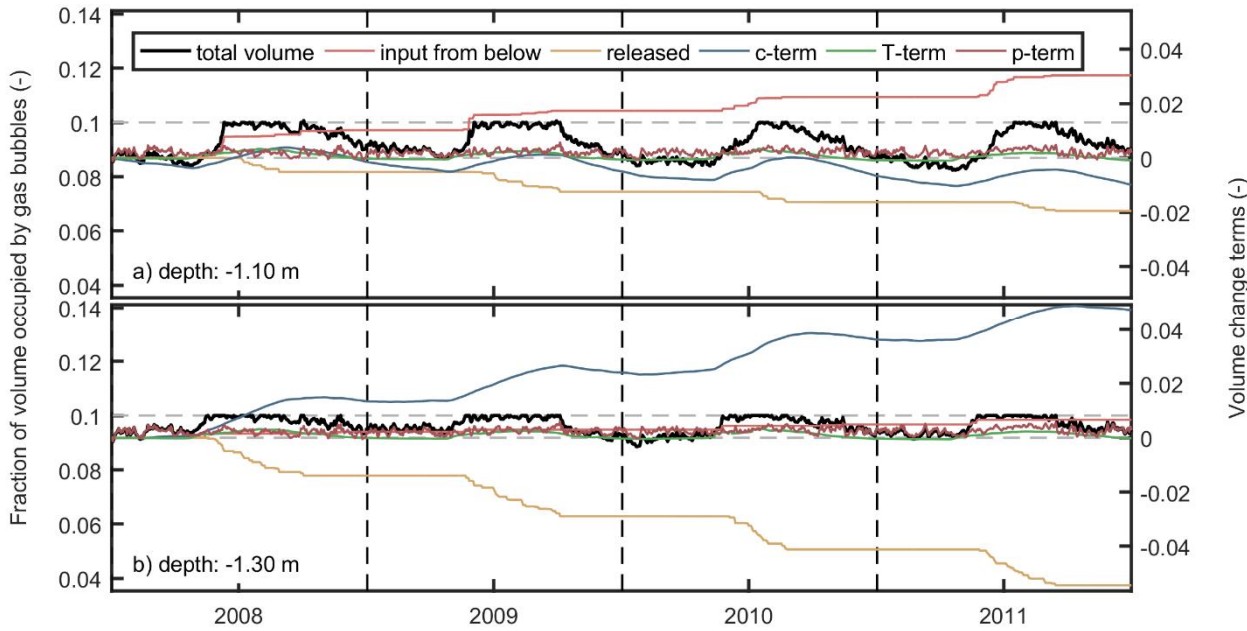

**Figure 4: Fraction of volume occupied by gas bubbles at two example depths calculated with the EBG approach. Black lines show the modelled volume (left y-axis) and the other continuous lines (right y-axis) show the terms causing the volume changes. Therefore sum of the lines in colour yield the changes in the black line at every time step. Light grey dashed line highlights the volume threshold ($f_{V\max}=0.1$), after which additional volume increase was released in an ebullition event.**

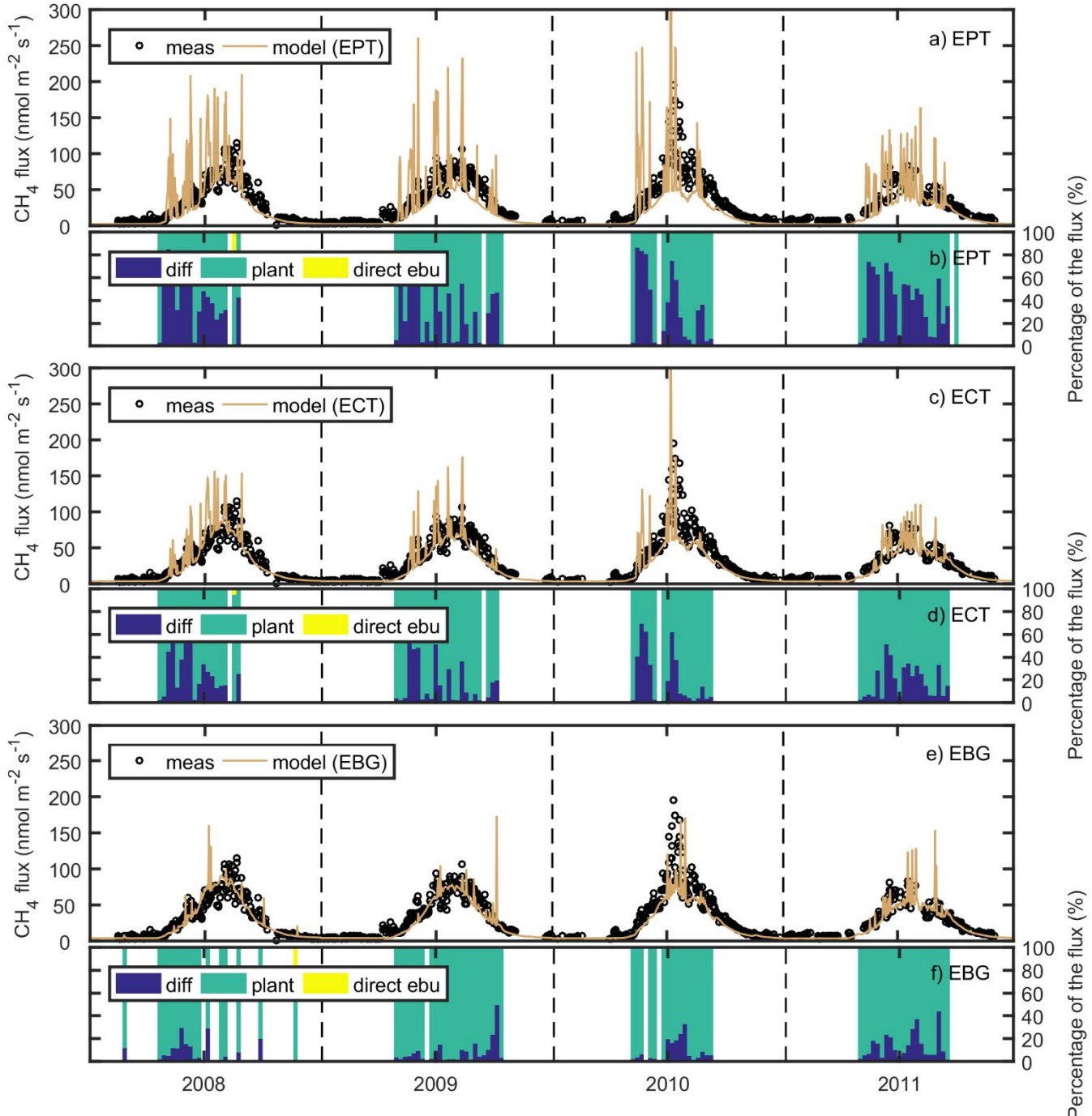

**Figure 5:Daily CH₄ fluxes from the fen to the atmosphere obtained using different approaches for modelling ebullition (subplots a, c and e) and weekly contribution of three emission pathways to flux (subplots b, d and f). Measured fluxes are shown with circles. diff = diffusion, plant = plant transport, direct ebu = ebullition directly to the surface. The diffusion flux contains the ebullition flux released to the lowest air layer which is usually below the peat surface. White areas in the subplots b, d and f correspond to periods when over 99 % of the flux was related to the plant-mediated transport.**

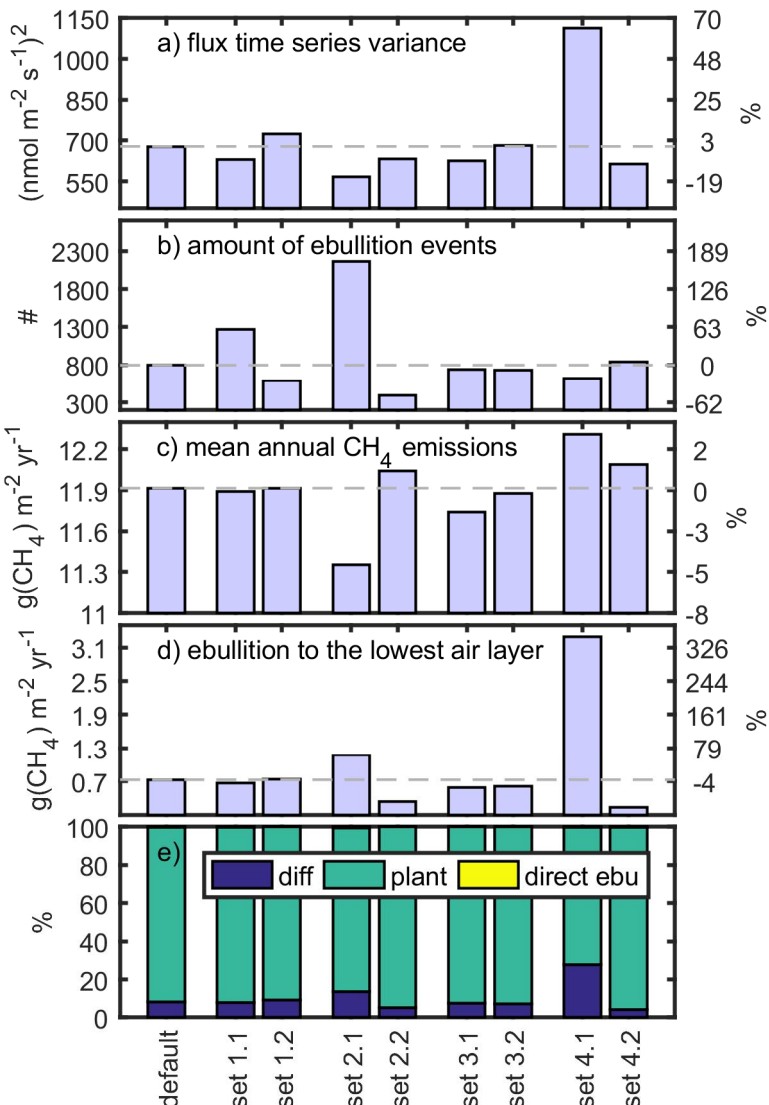

**Figure 6: Results from the EBG sensitivity analysis (see parameter values in Table 1): variance of the CH₄ flux time series (subplot a), total amount of ebullition events (b), mean annual CH4 emissions (c), mean annual ebullition flux to the lowest air layer (d) and the average contribution of different emission pathways to the overall CH4 flux to the atmosphere (e). Direct ebullition to the surface is negligible on annual scale and hence not visible in the subplot e. Absolute values are on the y-axis on the left and change relative to the default run are on the right side. Grey dashed lines highlight the values obtained with the default run.**

**Table 1: Parameters needed in the EBG ebullition module. Default values are given, along with different values used in the sensitivity analysis.**

| Parameter | Description | Default values | Parameter set 1.1 ($f_{Vmax}$ low) | Parameter set 1.2 ($f_{Vmax}$ high) | Parameter set 2.1 ($C$ low) | Parameter set 2.2 ($C$ high) | Parameter set 3.1 ($N_{bub}$ low) | Parameter set 3.2 ($N_{bub}$ high) | Parameter set 4.1 (P low) | Parameter set 4.2 (P high) |
|---|---|---|---|---|---|---|---|---|---|---|
| $f_{Vmax}$ (dimensionless) | Threshold fraction of pore space filled with gas bubbles needed for ebullition | 0.1 | 0.05 | 0.15 | 0.1 | 0.1 | 0.1 | 0.1 | 0.1 | 0.1 |
| $C$ (mol mol$^{-1}$) | CH$_4$ mixing ratio in the bubbles | 0.5 | 0.5 | 0.5 | 0.2 | 0.8 | 0.5 | 0.5 | 0.5 | 0.5 |
| $N_{bub}$ (#) | Amount of bubbles in one 0.2 m thick model layer | 100 | 100 | 100 | 100 | 100 | 10 | 1000 | 100 | 100 |
| P (dimensionless) | Probability that the released bubble volume will get stuck at one 0.2 m thick model layer while it is ascending | 0.3 | 0.3 | 0.3 | 0.3 | 0.3 | 0.3 | 0.3 | 0 | 0.5 |

**Table 2: Annual CH$_4$ emissions (g(CH$_4$) m$^{-2}$ yr$^{-1}$) estimated with HIMMELI model using different ebullition modules and measured with eddy covariance. Values in parentheses show the relative contribution (%) of each CH$_4$ emission pathway to the total annual emission estimate.**

|      |                                              | EBG       | ECT       | EPT      | Measured |
|------|----------------------------------------------|-----------|-----------|----------|----------|
| 2008 | Total                                        | 12.9      | 13.7      | 13.9     | 11.9     |
|      | Plant-transport                              | 12.2 (94) | 11.3 (82) | 9.0 (65) |          |
|      | Diffusion                                    | 0.7 (6)   | 2.4 (18)  | 4.7 (34) |          |
|      | Ebullition                                   | 0 (0)     | 0 (0)     | 0.1 (1)  |          |
|      | Ebullition to the lowest air filled pore space[a] | 0.5  | 2.4       | 4.8      |          |
| 2009 | Total                                        | 12.4      | 12.3      | 12.6     | 12.8     |
|      | Plant-transport                              | 11.5 (93) | 10.2 (83) | 8.0 (63) |          |
|      | Diffusion                                    | 0.8 (7)   | 2.1 (17)  | 4.6 (37) |          |
|      | Ebullition                                   | 0 (0)     | 0 (0)     | 0 (0)    |          |
|      | Ebullition to the lowest air filled pore space [a] | 0.6 | 1.9      | 4.5      |          |
| 2010 | Total                                        | 11.6      | 12.4      | 12.4     | 14.7     |
|      | Plant-transport                              | 10.6 (91) | 9.9 (80)  | 7.6 (61) |          |
|      | Diffusion                                    | 1.0 (9)   | 2.5 (20)  | 4.8 (39) |          |
|      | Ebullition                                   | 0 (0)     | 0 (0)     | 0 (0)    |          |
|      | Ebullition to the lowest air filled pore space [a] | 0.7 | 2.3      | 4.7      |          |
| 2011 | Total                                        | 10.8      | 10.7      | 10.7     | 11.6     |
|      | Plant-transport                              | 9.5 (87)  | 8.8 (82)  | 6.6 (61) |          |
|      | Diffusion                                    | 1.4 (13)  | 2.0 (18)  | 4.2 (39) |          |
|      | Ebullition                                   | 0 (0)     | 0 (0)     | 0 (0)    |          |
|      | Ebullition to the lowest air filled pore space [a] | 1.0 | 1.7      | 4.0      |          |

[a] Not included in the total annual CH$_4$ emission.