# Peer review of "Technical Note: Comparison of methane ebullition modelling approaches used in terrestrial wetland models"

_Biogeosciences, 2017_

## Referee Comment (RC1) · Prof Roulet (Referee) · 15 Aug 2017

Overall impression

This is an interesting paper. The problem of methane transport has vexed empiricists and modellers alike since the exchange of methane from wetlands and the atmosphere was identified as a serious problem. This manuscript presents three models of methane ebullition from peatlands and then assesses how they perform against annual totals and a time series of half-hourly methane fluxes. They show that all the models give approximately the same annual flux but the proportion for the emissions transported via bubbles varies among the models. They conclude, based on comparison

with half-hour eddy covariance data that the model they developed, the free-phase gas volume model (EBG), gives more realistic results than the other two models. I am not convinced by the evidence presented that the EBG more is better because I do not see how half-hour average EC fluxes are a good measure of the bubble flux. My concern is the boundary layer mixing blurs the bubble signal and that the post-processing removes concentrations from the high frequency data (e.g. 10 Hz) that indicate bubbles. I have wondered whether a comparison of the co-spectra of the momentum and concentrations could be used to identify the frequency of bubbles? Bubbles are very hard to measure well. In flooded wetlands funnel traps give a time integrated measure to the magnitude of the bubble flux but they do not work in peatlands. When auto-chambers are used one can see bubbles but examining the trace of concentration over time. At our site, Mer Bleue, we saw evidence of bubbles in less than 1% of our fluxes but in more fen systems such as Sallies fen (Goodrich in the references) bubbles appear in most chamber closers. Ideally the authors would have higher frequency records or used chambers where they can actually see the bubbles but it has to be an automated systems. Probably as good a test of the models is measures of the changes in methane storage. The authors point out this is not easy to do with interfering with the concentrations but diffusion samplers such as peppers or continuous flow samples could give evidence for the changes in storage – this is the state variable in all the models. I think there is value in this manuscript (see below) but I think the authors should outline the ways they think the models could be tested. This would help others recognize the data they are sitting on could be used as evidence to attempt to refute the models. It is only through this testing we will gain confidence in the models and differentiate which model is more appropriate.

I do think the manuscripts serves a useful purpose in setting out the three approaches to modelling ebullition. It is the clearest explanation of these kinds of models I have seen and on that basis alone I think it serves a very useful addition to the literature. On theoretical arguments I do think their EBG model stands up better than the other two models but the authors should be inviting or stimulating the community to test

the models. However we are still faced a significant problem with estimating methane production: Rprod in equation (1). It does not really matter how elegant our models of the transport mechanisms are if production is poorly estimated.

Minor points

Pg 1 Ln 25 Not sure of this number. Ebullition is important where it is important but when plants are present that root below the water table plant mediated is important. Diffusion can't be that large. Also think there is steady bubble flux. Regardless it does not matter because it is important.

Pg 1 Ln 30 This is an understatement which is why I question the 0 to 70% claim above.

Pg 5 Ln 21 Subscript 4 on CH4

Pg 10 Ln 11-13 This is fine but it my experience that most peatlands do not have significant confining layers. Beaver ponds are a good example - they a constantly bubbling and the bubbles are shallow.

Pg 10 Ln 35 Patrick Crill is measuring concentrations in peat with circulating diffusion samplers that equilibrate with soil concentrations. They would not sample bubbles but they would tell you the depth and duration that the concentrations approach and exceed saturation. Also peepers would tell you the same thing.

Pg 11 Ln 4 Yes but the easiest thing to measure is the concentration profiles in some non-destructive manner that does not require putting a negative pressure to extract the samples. My guess is diffusion samples circulating through a CH4 analyzer would be the best route to get at this problem. At least then you are measuring the temporal variability of the state variable. If the diffuser were of a sufficient length, say several meters, they could obtain spatial averages. What do you recommend as a test?

Pg 11 Ln 14-26 I am not sure how one can use EC to determine a bubble flux. What EBG shows is that it matches the temporal pattern of the EC fluxes but it can't see bubbles. I have always wondered in the high frequency data the concentration and

momentum spectra should see departures that would indicate bubbles being mixed in the boundary layer. Automated chambers see bubble events - the time trace of concentrations show step changes. You refer to Goodrich et al. And they saw thus at Sallies fen in NH.

Nigel Roulet, McGill University, Montreal, August 2017

---

## Referee Comment (RC2) · J. Ramirez (Referee) · 29 Aug 2017

Comments

This paper does address a subject that is relevant for BG and provides a good overview of existing methane ebullition models. The explanation of the models is clear and background information from the literature is provided. The aim of the paper is straightforward and the study intends to test ebullition models using observed data. Furthermore, the models are used to produce outputs that cannot be directly verified using observations. This is fine, and is one of the major points of modelling systems. What is unclear, is how the performance of the models are gauged against the observed data.

The main topic of this study is methane emissions via ebullition and the authors do not clearly describe how ebullition events are measured using Eddy Covariance (EC) data. In the end, the authors compare model output from all methane transport mechanisms combined (ebullition, plant mediated, and diffusion) against the EC data. Unfortunately, this comparison does not allow the authors to derive any definitive conclusions about modelling methane ebullition.

The paper can also benefit from more explanation on the collection and processing of the field data. On the modelling side, the authors should provide information on calibration of the models and provide reasons if calibration was not performed and how this could affect model results. A brief model sensitivity analysis is presented, and it would be interesting to expand this section with a more in depth, systematic sensitivity analysis that includes a figure. The majority of the paper is well written, but some sentences need restructuring (see comments below). To improve clarity the manuscript should be read by a native English speaker.

Technical corrections

Pg 1 Line 25: Ebullition is not important in all wetlands. perhaps change sentence by adding, "in some wetlands"

Pg 1 Line 28: Ebullition is not only sporadic in space, but also in time and you should provide background evidence for this (see introduction of Ramirez et al. [2017]).

Pg 1 Line 36: Total volume of what?

Pg 2 Line 8: Mention that both increasing and decreasing atmospheric pressure trigger ebullition and provide references.

Pg2 Line 19: Avoid starting sentences with a number, instead spell out the number. This sentence also needs to be restructured.

Pg2 Line 20: Replace questioned with questionable?

Pg 2 Line 28: Replace inflict with produce.

Pg3 Line 1: Is this upward transport, and/or also lateral transport? Be specific.

Pg3: Does the model have a spatial resolution? How many layers exist in the peat column and how thick are the layers? Introducing this information early on helps the reader visualise how the model operates.

Pg4 Line 8: Can you explain better what the is meant by the lowest air layer. Here you mention the model time step, can you provide the actual time step early in the model introduction (e.g. hrs or days).

Pg 4 Line 20: Again, the lowest air layer is vague, perhaps define it earlier. Replace ascend with ascent.

Pg 6 Line 9: You introduce a layer thickness 0.2 m, is this the case for all three models (see comments above regarding spatial resolution).

Pg7 Line 1: replace gapfilled with gap-filled

Pg 7 Line 4: Can you provide evidence in the literature that ebullition events from peat can be measured using EC.

Pg 7 Line 10: Can you provide further explanation as to how ebullition was detected using the EC data. Include details about the post-processing of the data.

Pg 7 Line 33: Avoid starting sentences with a number, instead spell out the number. Apply to other instances within the manuscript.

Figure 4: Dashed line is not clearly visible.

Pg 8 Line 36: Change "got stuck" and "alive" with formal words.

Pg 8 Line 38 and 39: "Inflicted" is not the correct word choice.

Figure 5a: It is difficult to distinguish between the four datasets plotted. Consider restructuring this figure with a separate panel for each model output, with the observed

data superimposed. Additional figures you may consider are histograms and scatter-plots between observed and modelled emissions (with the later the differences are clearly noticeable).

Pg 9 Line 4: Does figure 5a compare modelled and observed CH4 emissions considering all three transport mechanisms (diffusion, plant mediated, and ebullition)? If this is the case, it is quite confusing because the manuscript up to now was focusing on ebullition, and I was expecting a comparison between observed and modelled ebullition emissions. Please provide further explanation.

Pg 9 Line 10 and 11: This variability is difficult to see in Figure 5a, see comment above.

Pg 9 Line 14: Can you provide more explanation how the R2 were derived for each model.

Pg 9 Line 22: Replace the phrase "complicates the picture" with formal wording and explanation.

Pg 9 Section 3.4: This a model sensitivity section and should be presented before the results section.

Pg 10 Line 11-13: How do the models support the rupturing of confining layers if peat micro structure (e.g. woody layers) is not entirely represented within the models? Can you provide explanation for this conclusion.

Pg 11 Line 5: Sentence needs restructuring.

Pg 11 Line 23: You mention calibration of methane models performed in other studies, can you explain why you chose not to perform a calibration prior to model testing and how this affects your results?

Pg 11 Discussion section: Assuming that you cannot definitively identify ebullition events in the EC data, it is possible that the EC data contains few ebullition events. Could this further explain the mismatch between observed and modelled CH4 emissions? Please address this possibility in the discussion section.

References

Ramirez, J., A. Baird, and T. Coulthard (2017), The effect of sampling effort on estimates of methane ebullition from peat, Water Resour. Res.
* * *

---

## Author Comment (AC1) · 14 Nov 2017

**AUTHOR RESPONSES TO REVIEWER COMMENTS**

Title: Technical Note: Comparison of methane ebullition modelling approaches used in terrestrial wetland models
Author(s): Olli Peltola et al.
MS No.: bg-2017-274
MS Type: Technical note

We thank the reviewers for their constructive criticism and thorough comments which helped improve the manuscript. The main criticism from both reviewers was directed towards the comparison of modelling results against eddy covariance (EC) data. They argued that EC fluxes are not a good measure of the bubble flux and hence are not ideal for this kind of study. We partly agree with this notion and will modify the text accordingly. Please see more details below.

The reviewer comments are addressed individually below. The comments are shown in bold and responses with regular font.

On behalf of the authors,
Olli Peltola

REFEREE #1: Prof. Nigel Roulet

Overall impression
This is an interesting paper. The problem of methane transport has vexed empiricists and modellers alike since the exchange of methane from wetlands and the atmosphere was identified as a serious problem. This manuscript presents three models of methane ebullition from peatlands and then assesses how they perform against annual totals and a time series of half-hourly methane fluxes. They show that all the models give approximately the same annual flux but the proportion for the emissions transported via bubbles varies among the models. They conclude, based on comparison with half-hour eddy covariance data that the model they developed, the free-phase gas volume model (EBG), gives more realistic results than the other two models. I am not convinced by the evidence presented that the EBG more is better because I do not see how half-hour average EC fluxes are a good measure of the bubble flux. My concern is the boundary layer mixing blurs the bubble signal and that the post-processing removes concentrations from the high frequency data (e.g. 10 Hz) that indicate bubbles. I have wondered whether a comparison of the co-spectra of the momentum and concentrations could be used to identify the frequency of bubbles? Bubbles are very hard to measure well. In flooded wetlands funnel traps give a time integrated measure to the magnitude of the bubble flux but they do not work in peatlands. When auto-chambers are used one can see bubbles but examining the trace of concentration over time. At our site, Mer Bleue, we saw evidence of bubbles in less than 1% of our fluxes but in more fen systems such as Sallies fen (Goodrich in the references) bubbles appear in most chamber closers. Ideally the authors would have higher frequency records or used chambers where they can actually see the bubbles but it has to be an automated systems. Probably as good a test of the models is measures of the changes in methane storage. The authors point out this is not easy to do with interfering with the concentrations but diffusion samplers such as peppers or continuous flow samples could

give evidence for the changes in storage – this is the state variable in all the models. I think there is value in this manuscript (see below) but I think the authors should outline the ways they think the models could be tested. This would help others recognize the data they are sitting on could be used as evidence to attempt to refute the models. It is only through this testing we will gain confidence in the models and differentiate which model is more appropriate. I do think the manuscripts serves a useful purpose in setting out the three approaches to modelling ebullition. It is the clearest explanation of these kinds of models I have seen and on that basis alone I think it serves a very useful addition to the literature. On theoretical arguments I do think their EBG model stands up better than the other two models but the authors should be inviting or stimulating the community to test the models. However we are still faced a significant problem with estimating methane production: Rprod in equation (1). It does not really matter how elegant our models of the transport mechanisms are if production is poorly estimated.

RESPONSE: We want to thank Prof. Roulet for his comments. We do agree with him that the boundary layer mixing does blur the ebullition signal in the fluxes derived using the eddy covariance (EC) technique, yet it still does capture the methane ($CH_4$) flux caused by the released bubbles accurately. We argue that the ebullition signal can be seen in the 10-Hz $CH_4$ concentration time series as events with elevated $CH_4$ concentrations. These events are related to air parcels carrying the $CH_4$ released by bubbles and transported from the surface upwards and eventually past the EC measurement devices which then record these $CH_4$ emission bursts. We carefully post-processed the data in order to retain these events throughout the EC data processing chain. In fact, we are currently investigating whether these events could be separated from the turbulent time series using e.g. wavelet transforms to derive ecosystem scale bubble flux (topic of another study).

Another way to frame the problem raised by the reviewer is that the EC system estimates the fluxes at ecosystem scale whereas the model estimates the $CH_4$ fluxes from a single peat column. Hence there is an obvious mismatch in horizontal scale. EC source area contains locations where ebullition is taking place and locations where ebullition is at the same time absent. The conventionally estimated 30-min averaged EC fluxes combine all the different emission pathways (diffusion, plant transport and ebullition) at ecosystem scale and hence the modelled and measured fluxes are not fully comparable. Therefore, we will modify the manuscript so that we will mention this scale mismatch in all relevant locations (abstract, Sect. 3.3 & 4). We will emphasize more the comparison between the ebullition modelling methods instead of the validation against EC measurements. As suggested by the reviewer, we will also add a small section to Discussion where we will mention the more appropriate datasets against which this kind of models should be tested (auto-chambers, changes in $CH_4$ stored in the peat).

Minor points
Pg 1 Ln 25 Not sure of this number. Ebullition is important where it is important but when plants are present that root below the water table plant mediated is important. Diffusion can't be that large. Also think there is steady bubble flux. Regardless it does not matter because it is important.

RESPONSE: Agreed. We will remove the percentages and simply just mention that in some studies ebullition has been observed to contribute a significant fraction to the overall $CH_4$ flux to the atmosphere.

Pg 1 Ln 30 This is an understatement which is why I question the 0 to 70% claim above.

RESPONSE: We will emphasize the lack of knowledge in this matter more.

Pg 5 Ln 21 Subscript 4 on $CH_4$

RESPONSE: Thanks, will be corrected.

Pg 10 Ln 11-13 This is fine but it my experience that most peatlands do not have significant confining layers. Beaver ponds are a good example - they a constantly bubbling and the bubbles are shallow.

RESPONSE: This is a good point and many recent studies have shown that the bubbles are mostly formed close to the surface, as already mentioned later on in the Discussion.

Pg 10 Ln 35 Patrick Crill is measuring concentrations in peat with circulating diffusion samplers that equilibrate with soil concentrations. They would not sample bubbles but they would tell you the depth and duration that the concentrations approach and exceed saturation. Also peepers would tell you the same thing.

RESPONSE: We are using similar diffusion samplers at our lake site to estimate $CO_2$ concentration profiles in the lake. The plan is to set up similar system at our peatland site to measure $CH_4$ concentrations in the peat. However, such measurements average the soil pore water concentrations over a relatively large area, since long sampling tubes are needed before the air in the tubes is in equilibrium with the surrounding pore water concentrations (diffusion through the tube walls is slow). Hence, most likely the small scale variability in the pore water concentration cannot be captured with such systems. This variability might be crucial for creating favorable conditions for bubble formation at depths where on average low pore water concentrations inhibit the bubble formation. These issues are already mentioned in the Discussion section of the manuscript.

Pg 11 Ln 4 Yes but the easiest thing to measure is the concentration profiles in some non-destructive manner that does not require putting a negative pressure to extract the samples. My guess is diffusion samples circulating through a $CH_4$ analyzer would be the best route to get at this problem. At least then you are measuring the temporal variability of the state variable. If the diffuser were of a sufficient length, say several meters, they could obtain spatial averages. What do you recommend as a test?

RESPONSE: We are not sure what the referee is meaning with this comment, since the text on page 11, line 4 does not discuss the concentration profiles. Maybe due to an oversight this comment is refers to wrong location in the manuscript. In any case, we agree with the referee that the diffusion samplers are a convenient way to measure concentrations in the water, however they cannot capture the small scale variability in the pore water concentrations which could be vital for bubble formation. Moreover, we agree that the pore water concentration is the state variable against which in general this kind of models should be ideally tested.

Pg 11 Ln 14-26 I am not sure how one can use EC to determine a bubble flux. What EBG shows is that it matches the temporal pattern of the EC fluxes but it can't see bubbles. I have always wondered in the high frequency data the concentration and momentum spectra should see departures that would indicate bubbles being mixed in the boundary layer. Automated chambers see bubble events - the time trace of concentrations show step changes. You refer to Goodrich et al. And they saw thus at Sallies fen in NH.

RESPONSE: As mentioned above, we argue that the bubble flux can be seen in the 10-Hz $CH_4$ concentration time series as events with significantly elevated $CH_4$ concentrations. These events contribute to the covariance between the vertical wind speed and the gas concentrations as any other fluctuation in the time series and hence EC does capture the ebullition signal. However, it is just mixed with all the other emission pathways taking place within the EC ecosystem-scale source area whereas the model works on a single peat column scale. As mentioned before, we will emphasize this point more in the revised version of the manuscript.

Spectral analysis is not necessarily the best tool for detecting these events as the spectra are calculated from the complete time series which combines all the emission signals. Instead, wavelets and wavelet transforms allow separating individual events from the high frequency time series and hence possibly enable the estimation of ecosystem scale bubble flux. We are currently investigating whether the ebullition signal could be disentangled from the 10-Hz time series using wavelets, but that is a topic for another study.

REFEREE # 2: Dr. Jorge Remirez

Comments
This paper does address a subject that is relevant for BG and provides a good overview of existing methane ebullition models. The explanation of the models is clear and background information from the literature is provided. The aim of the paper is straightforward and the study intends to test ebullition models using observed data. Furthermore, the models are used to produce outputs that cannot be directly verified using observations. This is fine, and is one of the major points of modelling systems. What is unclear, is how the performance of the models are gauged against the observed data.
The main topic of this study is methane emissions via ebullition and the authors do not clearly describe how ebullition events are measured using Eddy Covariance (EC) data. In the end, the authors compare model output from all methane transport mechanisms combined (ebullition, plant mediated, and diffusion) against the EC data. Unfortunately, this comparison does not allow the authors to derive any definitive conclusions about modelling methane ebullition. The paper can also benefit from more explanation on the collection and processing of the field data. On the modelling side, the authors should provide information on calibration of the models and provide reasons if calibration was not performed and how this could affect model results. A brief model sensitivity analysis is presented, and it would be interesting to expand this section with a more in depth, systematic sensitivity analysis that includes a figure. The majority of the paper is well written, but some sentences need restructuring (see comments below). To improve clarity the manuscript should be read by a native English speaker.

RESPONSE: We thank Dr. Ramirez for his comments. As already discussed above, we acknowledge the shortcomings of EC data and will revise the manuscript accordingly by mentioning the spatial scale mismatch in all relevant locations and shifting the focus of the manuscript more into the modelling approach comparison. We will also add more details about the measurement setup and data processing to the Sect. 2.2, however this is a modelling study and hence we would like to keep this section as short as possible.

The HIMMELI model was calibrated in Susiluoto et al. (2017) and this will be mentioned in the Sect. 2.1. However, we would like to emphasize that the main aim of this study is to compare different ebullition modelling approaches, not to model the $CH_4$ emissions from the peatland with high accuracy. We argue that for this kind of comparison model calibration is not crucial, albeit in general for $CH_4$ cycling modelling it should be done with care.

The parameters used in the three ebullition modelling approaches were not calibrated, instead typical values found in the literature were used. We opted for this approach since the model parameters are often correlated with each other (e.g. Susiluoto et al., 2017), making careful calibration time-consuming and sensitive process. Furthermore, full model calibration with different ebullition approaches would make it difficult to evaluate whether the differences in modelling results stem from ebullition modelling approach or from the model calibration. Hence, we believe that using parameter values found from the literature should provide robust results for the readers to gauge the differences between the ebullition modelling approaches. We will modify

the manuscript and try to make it clearer that parameter values from the literature are used, not calibrated ones.

As requested by the reviewer, we will expand the model sensitivity section of the study and include an appropriate figure detailing the most relevant results from this analysis.

Technical corrections
Pg 1 Line 25: Ebullition is not important in all wetlands. perhaps change sentence by adding, "in some wetlands"

RESPONSE: Thanks, we will modify this part.

Pg 1 Line 28: Ebullition is not only sporadic in space, but also in time and you should provide background evidence for this (see introduction of Ramirez et al. [2017]).

RESPONSE:In this sentence the word "events" is supposed to emphasize that ebullition is also sporadic in time. We will modify this sentence so that it is clearer that the temporal variability is also significant and add an appropriate reference.

Pg 1 Line 36: Total volume of what?

RESPONSE: Of the peat volume (see Rosenberry et al., 2006). We will modify this sentence so that this will be clear.

Pg 2 Line 8: Mention that both increasing and decreasing atmospheric pressure trigger ebullition and provide references.

RESPONSE: Increasing pressure is already mentioned on page 2 line 11 and a reference is also provided.

Pg2 Line 19: Avoid starting sentences with a number, instead spell out the number. This sentence also needs to be restructured.

RESPONSE: Yes, we agree that sentences should not begin with a number. We will fix this part.

Pg2 Line 20: Replace questioned with questionable?

RESPONSE: The sentence here means that this modelling approach is controversial given the current knowledge of the ebullition process. Hence we argue that "questioned" is correct.

Pg 2 Line 28: Replace inflict with produce.

RESPONSE: Yes, produce sounds better.

Pg3 Line 1: Is this upward transport, and/or also lateral transport? Be specific.

RESPONSE: Thanks, we will specify that it is upward transport.

Pg3: Does the model have a spatial resolution? How many layers exist in the peat column and how thick are the layers? Introducing this information early on helps the reader visualise how the model operates.

RESPONSE: The model is a 1D-column model and hence does not as such have a spatial resolution. We run the model with 2 m of peat with ten layers, therefore each layer was 0.2 m thick. We will add these pieces of information to the Sect. 2.1.

Pg4 Line 8: Can you explain better what the is meant by the lowest air layer. Here you mention the model time step, can you provide the actual time step early in the model introduction (e.g. hrs or days).

RESPONSE: The lowest air layer is the model layer which is the lowest model layer which is above water level and below peat level. The model output time step is one day, but the model uses a shorter internal time step in order to ensure numerical stability. We will mention these details in the model description section (Sect. 2.1.).

Pg 4 Line 20: Again, the lowest air layer is vague, perhaps define it earlier. Replace ascend with ascent.

RESPONSE: We will define the lowest air layer in Sect. 2.1. Thanks, we will replace ascend with ascent.

Pg 6 Line 9: You introduce a layer thickness 0.2 m, is this the case for all three models (see comments above regarding spatial resolution).

RESPONSE: Yes, the model settings were kept constant and only the ebullition modelling approach was varied in this study. We will mention the layer thickness in Sect. 2.1.

Pg7 Line 1: replace gapfilled with gap-filled

RESPONSE: Thanks, will be replaced.

Pg 7 Line 4: Can you provide evidence in the literature that ebullition events from peat can be measured using EC.

RESPONSE: At least Hargreaves et al. (2001) and Sachs et al. (2008) argue that their EC $CH_4$ flux measurements were significantly affected by ebullition. However, a study where ebullition is recorded with multiple independent measurement systems (e.g. chambers and EC) at the same is yet to be published.

Pg 7 Line 10: Can you provide further explanation as to how ebullition was detected using the EC data. Include details about the post-processing of the data.

RESPONSE: We argue that the ebullition events can be seen in the 10 Hz EC data as events with significantly elevated $CH_4$ concentrations. These fluctuations in the concentration time series contribute to the flux the same way as any other fluctuation. However, a problem arises from the scale mismatch between EC and the model results. See the discussion above. During data post-processing we made sure that these events were not flagged as erroneous measurements, e.g. by de-spiking. We will add details about data post-processing and mention the scale mismatch in all relevant locations.

Pg 7 Line 33: Avoid starting sentences with a number, instead spell out the number. Apply to other instances within the manuscript.

RESPONSE: Thanks, we will remove numbers from the beginning of sentences.

Figure 4: Dashed line is not clearly visible.

RESPONSE: We will plot the dashed lines with darker colors.

Pg 8 Line 36: Change "got stuck" and "alive" with formal words.

RESPONSE: We will reformulate the sentence as: "Bubble volumes released from deeper layers and which were re-attached during their ascent maintained the gas volume at -1.1 m depth even though the c-term on average decreased the volume (cf. Fig 4a)."

Pg 8 Line 38 and 39: "Inflicted" is not the correct word choice.

RESPONSE: We will replace "inflicted" with "generated".

Figure 5a: It is difficult to distinguish between the four datasets plotted. Consider restructuring this figure with a separate panel for each model output, with the observed data superimposed. Additional figures you may consider are histograms and scatterplots between observed and modelled emissions (with the later the differences are clearly noticeable).

RESPONSE: We decided to plot all the lines to the same panel, since we wanted to have also the different emissions pathways in the same figure (subplots b-d). However, based on this comment we will modify the figure so that it will have six panels: one for comparing against measurements and one for showing the different emissions pathways for each ebullition modelling approach.

We will not add the additional figures suggested by the reviewer (histograms or scatter plots), due to the scale mismatch between model results and EC derived $CH_4$ fluxes (see the discussion above). Moreover, in the revised manuscript we will emphasize more the comparison between modelling approaches and less the validation against EC measurements.

Pg 9 Line 4: Does figure 5a compare modelled and observed $CH_4$ emissions considering all three transport mechanisms (diffusion, plant mediated, and ebullition)? If this is the case, it is quite confusing because the manuscript up to now was focusing on ebullition, and I was expecting a comparison between observed and modelled ebullition emissions. Please provide further explanation.

RESPONSE: Yes, it includes emissions via all three emission pathways. There are at least three reasons for this:
1) In the HIMMELI model bubbles released to lowest air layer are not recorded as ebullition flux since diffusion is still needed to transport the released $CH_4$ to the atmosphere. Hence the diffusion flux contains the ebullition flux released to the lowest air layer which is usually below the peat surface. Therefore ebullition flux cannot be completely separated from the diffusion flux. This is also mentioned in the Fig. 5 caption.
2) EC measurements contain all the three emission pathways and there is no robust method to separate the emissions pathways from each other.
3) The ebullition modelling approach alters the model $CH_4$ cycling in a fundamental way (see e.g. the $CH_4$ concentration profiles in Fig. 2 in the manuscript) and hence also the other emission pathways are affected. For instance the magnitude of plant transport is quite different between the EPT and the EBG

approach (see Table 2 in the manuscript). Hence we argue that it would not be sufficient to compare only the ebullition flux against measurements.

Pg 9 Line 10 and 11: This variability is difficult to see in Figure 5a, see comment above.

RESPONSE: We will modify the figure, see above.

Pg 9 Line 14: Can you provide more explanation how the R2 were derived for eachmodel.

RESPONSE: Thanks for this comment, since it allowed us to find an oversight in our analysis. Previously the $R^2$ value was calculated as the squared Pearson correlation coefficient, however a more appropriate way to estimate the coefficient of determination ($R^2$) in our case would be:

$$R^2 = 1 - \frac{\sum_i (y_i - f_i)^2}{\sum_i (y_i - \bar{y})^2},$$
(1)

where $y$ are the measurements and $f$ the modelled $CH_4$ fluxes. The overbar denotes averaging and subscript $i$ a running index over the whole time series. The equation above produces the following $R^2$ values for each model run: 0.63 for EBG, 0.56 for ECT and 0.35 for EPT. We will update these numbers to the manuscript.

Pg 9 Line 22: Replace the phrase "complicates the picture" with formal wording and explanation.

RESPONSE: We will reformulate this sentence as "although $CH_4$ oxidation in the peat column might be different between the modelling approaches".

Pg 9 Section 3.4: This a model sensitivity section and should be presented before the results section.

RESPONSE: We would like to keep the sensitivity section here, since at this point the reader is already familiar with the different concepts used in this study (for instance ebullition event and its profile). Hence it is easier for the reader to understand the differences between the sensitivity runs (see Table 1) without an elaborate explanation of the different concepts.

We want to emphasize that the default parameter values for the EBG approach were derived from literature and not from this sensitivity analysis. This section is meant merely for future studies using the EBG approach to understand which parameters are the most crucial for this modelling approach.

Pg 10 Line 11-13: How do the models support the rupturing of confining layers if peat micro structure (e.g. woody layers) is not entirely represented within the models? Can you provide explanation for this conclusion.

RESPONSE: This conclusion is based on the depth of the ebullition events (mostly below 1 m, see Fig. 1) and on the temporal variability of the modelled ebullition flux, which follows from the threshold logic (concentration, pressure or volume) used in all approaches (ECT, EPT and EBG, respectively). The reviewer is correct that the model does not have as such a detailed description of peat microstructure and we will mention this at the end of the sentence which ends on line 13:

"Hence the results resemble the "deep peat" hypothesis put forward by Glaser et al. (2004) in which the free-phase gas is produced in deep (> 1 m) peat and trapped under semiconfined layers, even though peat micro structure (e.g. woody layers) was not described in the model."

Pg 11 Line 5: Sentence needs restructuring.

RESPONSE: The sentence will be replaced with sentence: "Clearly the effect sedges on ebullition should be studied more prior to these effects can be implemented in terrestrial $CH_4$ models".

Pg 11 Line 23: You mention calibration of methane models performed in other studies, can you explain why you chose not to perform a calibration prior to model testing and how this affects your results?

RESPONSE: As mentioned above, the HIMMELI model was already calibrated by Susiluoto et al. (2017) and this will be mentioned in Sect 2.1. In general, model calibration is a large task worth its own publication and was not included here. The ebullition modelling approaches were run using parameter values found from the literature, which has at least two advantages:
1) The differences between modelling results do not follow from differences in the model calibration. Model parameters are often correlated with each other (see e.g. Susiluoto et al., 2017) and hence if the model would have been calibrated separately for each ebullition modelling approach, then also other parameters not directly related to ebullition (e.g. plant transport related parameters) might change. Then it would be difficult to evaluate whether the differences between model runs would be related to differences between ebullition modelling approaches or to model calibration.
2) When using the literature values, the modelling results can be directly compared against other studies using the same parameter values. For instance ECT results are comparable with Wania et al. (2010) ebullition modelling, since similar approach with the same parameter values were used.

Pg 11 Discussion section: Assuming that you cannot definitively identify ebullition events in the EC data, it is possible that the EC data contains few ebullition events. Could this further explain the mismatch between observed and modelled $CH_4$ emissions? Please address this possibility in the discussion section.

RESPONSE: Yes, we agree that this is a possibility. We argue that this is related to the scale mismatch: the model estimates fluxes from a single peat column, whereas EC measures fluxes at ecosystem scale. The EC source area may contain locations where ebullition is taking place and locations where at the same it is absent. Therefore, the ebullition events are diluted in the conventionally calculated EC fluxes, yet their impact on ecosystem scale fluxes is captured accurately. As mentioned above, we will mention this in all relevant locations of the manuscript.

References
Ramirez, J., A. Baird, and T. Coulthard (2017), The effect of sampling effort on estimates of methane ebullition from peat, Water Resour. Res.

REFERENCES

Hargreaves, K. J., Fowler, D., Pitcairn, C. E. R., and Aurela, M.: Annual methane emission from Finnish mires estimated from eddy covariance campaign measurements, Theor Appl Climatol, 70, 203-213, 10.1007/s007040170015, 2001.

Rosenberry, D. O., Glaser, P. H., and Siegel, D. I.: The hydrology of northern peatlands as affected by biogenic gas: current developments and research needs, Hydrological Processes, 20, 3601-3610, 10.1002/hyp.6377, 2006.

Sachs, T., Wille, C., Boike, J., and Kutzbach, L.: Environmental controls on ecosystem-scale $CH_4$ emission from polygonal tundra in the Lena River Delta, Siberia, Journal of Geophysical Research: Biogeosciences, 113, n/a-n/a, 10.1029/2007JG000505, 2008.

Susiluoto, J., Raivonen, M., Backman, L., Laine, M., Mäkelä, J., Peltola, O., Vesala, T., and Aalto, T.: Calibrating a wetland methane emission model with hierarchical modeling and adaptive MCMC, Geosci. Model Dev. Discuss., 2017, 1-50, 10.5194/gmd-2017-66, 2017.

Wania, R., Ross, I., and Prentice, I. C.: Implementation and evaluation of a new methane model within a dynamic global vegetation model: LPJ-WHyMe v1.3.1, Geosci. Model Dev., 3, 565-584, 10.5194/gmd-3-565-2010, 2010.

---

## Referee Report (RR1)

Page 2, Line 13: add "of" between "effect temperature"

Page 2, Line 42: replace "the coming" with "future"

Page 4, Line 4: remove "as" in "called as the lowest…"

Page 10, Lines 29-33: Please improve clarity of this sentence and provide better terminology for "get stuck"

Page 10, Line 38: Provide better terminology for "not stuck"

Figure 5: Move legends for weekly contribution of three emission pathways because they obscure the plotted data